# Methazolamide Reduces the AQP5 mRNA Expression and Immune Cell Migration—A New Potential Drug in Sepsis Therapy?

**DOI:** 10.3390/ijms25010610

**Published:** 2024-01-03

**Authors:** Katharina Rump, Björn Koos, Dominik Ziehe, Patrick Thon, Tim Rahmel, Lars Palmowski, Britta Marko, Alexander Wolf, Andrea Witowski, Zainab Bazzi, Maha Bazzi, Jennifer Orlowski, Michael Adamzik, Lars Bergmann, Matthias Unterberg

**Affiliations:** Klinik für Anästhesiologie, Intensivmedizin und Schmerztherapie, Universitätsklinikum Knappschaftskrankenhaus Bochum, 44892 Bochum, Germany; bjoern.koos@rub.de (B.K.); dominik.ziehe@rub.de (D.Z.); patrick.thon@rub.de (P.T.); tim.rahmel@rub.de (T.R.); lars.palmowski@kk-bochum.de (L.P.); britta.marko@kk-bochum.de (B.M.); alexander.wolf@kk-bochum.de (A.W.); andrea.witowski@kk-bochum.de (A.W.); zainab.bazzi@rub.de (Z.B.); maha.bazzi@rub.de (M.B.); jennifer.orlowski@rub.de (J.O.); michael.adamzik@kk-bochum.de (M.A.); matthias.unterberg@kk-bochum.de (M.U.)

**Keywords:** aquaporins, *AQP5*, sepsis, sulfonamides, methazolamide, furosemide, dorzolamide, LPS, immune cell migration in sepsis, AQP5 inhibition

## Abstract

Sepsis is a life-threatening condition caused by the dysregulated host response to infection. Novel therapeutic options are urgently needed and aquaporin inhibitors could suffice as aquaporin 5 (*Aqp5*) knockdown provided enhanced sepsis survival in a murine sepsis model. Potential AQP5 inhibitors provide sulfonamides and their derivatives. In this study, we tested the hypothesis that sulfonamides reduce AQP5 expression in different conditions. The impact of sulfonamides on AQP5 expression and immune cell migration was examined in cell lines REH and RAW 264.7 by qPCR, Western blot and migration assay. Subsequently, whether furosemide and methazolamide are capable of reducing AQP5 expression after LPS incubation was investigated in whole blood samples of healthy volunteers. Incubation with methazolamide (10^−5^ M) and furosemide (10^−6^ M) reduced *AQP5* mRNA and protein expression by about 30% in REH cells. Pre-incubation of the cells with methazolamide reduced cell migration towards SDF1-α compared to non-preincubated cells to control level. Pre-incubation with methazolamide in PBMCs led to a reduction in LPS-induced *AQP5* expression compared to control levels, while furosemide failed to reduce it. Methazolamide appears to reduce *AQP5* expression and migration of immune cells. However, after LPS administration, the reduction in *AQP5* expression by methazolamide is no longer possible. Hence, our study indicates that methazolamide is capable of reducing *AQP5* expression and has the potential to be used in sepsis prophylaxis.

## 1. Introduction

Sepsis is an acute organ dysfunction caused by a dysregulated immune response to an infection [1]. Despite intense research in recent decades, sepsis is still one of the most common causes of death in industrialized nations [2] and causes millions of deaths every year [3]. It is assumed that there are 48.9 million sepsis cases per year worldwide, which are responsible for up to 20% of global deaths [4]. In addition, sepsis is one of the most expensive conditions treated in hospital [5]. 

Therapeutic options to specifically treat the dysregulated immune response in sepsis have been largely unsuccessful so far, and research is focused on identifying new key proteins and deriving therapeutic options [6]. A promising candidate is the water, hydrogen peroxide and CO_2_ channel aquaporin 5 (AQP5) as its expression appears to affect sepsis survival in patients and *Aqp5*-deficient mice are more likely to survive sepsis and have reduced immune cell migration after lipopolysaccharide (LPS)-induced inflammation compared to wild-type mice [7,8,9].

Furthermore, the C-allele of the *AQP5* A(-1364)C-promoter polymorphism is not only associated with a lower AQP5 expression [7] but also with reduced neutrophil migration and reduced 30-day mortality in severe sepsis [8].

The inhibition of AQP5 expression could, thus, become a new therapeutic approach in sepsis therapy [10]. Potential AQP5 inhibitors including some sulfonamides and their derivatives were described to have inhibitory effects on other aquaporins (AQPs), such as AQP4 [11] or AQP1 in different cells and tissues [12]. A beneficial effect of the carbonic anhydrase inhibitor methazolamide was demonstrated, as it is capable of shortening the duration of mechanical ventilation in a cohort comprising sepsis patients. This observation did not provide any physiological insights [13]. In our study, we chose furosemide as an established inhibitor of AQPs as well as methazolamide and dorzolamide, which are further developments of the established AQP5 inhibitor acetazolamide [12]. However, it is unclear whether sulfonamides inhibit LPS-induced AQP5 expression and, thereby, inhibit immune cell migration. In addition, it has to be considered that there may be a primary antimicrobial effect that is potentially beneficial in the treatment of sepsis, as sulfonamides represent a large family of synthetic antibiotics that have been known for a long time [14]. 

In addition, it is unclear at which point these potential AQP5 inhibitors have to be administered to reduce AQP5 expression—before or after LPS incubation. Therefore, the following hypotheses were tested in this work: (1) sulfonamides reduce AQP5 expression in the cell line REH. (2) Reduction in AQP5 expression by sulfonamides impacts REH-cell migration. (3) Pre-incubation with sulfonamides reduces LPS-induced AQP5 expression. (4) Sulfonamides reduce cytokine release after LPS incubation. 

## 2. Results

### 2.1. Effects of Different Concentrations of Sulfonamides on Cell Viability 

Firstly, the nontoxic concentration of sulfonamides was determined using a viability assay after the incubation of two cell lines with different concentrations of the agents for 24 h. The cell viability of REH cells was significantly reduced by about 21% by incubation with 10^−4^ M methazolamide (*p* < 0.0143; Figure 1A) and by about 8% with 10^−5^ M methazolamide (*p* = n.s.), while furosemide (10% reduction) and dorzolamide (17% reduction) only slightly reduced cell viability at a concentration of 10^−4^ M (*p* = n.s.; Figure 1B,C). Lower concentrations did not impact cell viability.

An analysis of cell viability in the murine RAW 264.7 cell line showed similar results. A concentration of 10^−3^ M methazolamide and furosemide significantly reduced cell viability by approximately 73% for methazolamide and 65% for furosemide (*p* < 0.0001; Figure 2A,B), while the other concentrations had no effect on RAW 264.7 cell viability.

### 2.2. The Effect of Sulfonamides on AQP5 Expression in the Human Cell Line REH

The *AQP5* mRNA expression was examined in the cell line REH. Incubation with methazolamide at a concentration of 10^−5^ M reduced the *AQP5* expression compared to the control (*p* = 0.0418; Figure 3A) by about 20%. Furthermore, a concentration of 10^−6^ M furosemide reduced the *AQP5* expression (*p* = 0.0327; Figure 3B) by about 40%, while dorzolamide had no significant effect on the *AQP5* expression in REH cells (Figure 3C). 

Further down to the protein level, the incubation of the B-lymphocyte cell line REH with methazolamide led to a reduction in AQP5 protein expression after 24 h, while the protein expression remained the same under furosemide compared to the control condition (Figure 4A, n = 3). The AQP5 protein expression was compared to ACTB protein levels and the relative intensity of the control column was 1.2577 ± 0.0227, 0.6107 ± 0.0348 for methazolamide-treated cells and 1.2649 ± 0.0762 for furosemide-treated cells (*p* = 0.00151; ANOVA). In immunofluorescence staining furosemide caused a decrease in AQP5 distribution over the whole cell, while in methazolamide-treated cells, an increase in AQP5 signals was seen at the edge of the cell (Figure 4B). 

### 2.3. Methazolamide Reduces REH Cell Migration 

The REH cells showed targeted migration towards SDF1-α (*p* = 0.0193; Figure 5), which was seen by an increase in migration of about 18%. Pre-incubation of the cells with methazolamide reduced cell migration towards SDF1-α by about 10% compared to non-preincubated cells (*p* = 0.0473; Figure 5) to the control level. Furosemide only reduced cell migration by 3%, which can be stated to have no effect on REH cell migration (*p* = n.s.; Figure 5). 

### 2.4. Methazolamide Reduces AQP5 Expression in Peripheral Blood Mononuclear Cells (PBMCs)

We also examined *AQP5* expression in PBMCs of healthy donors after LPS application, and before and after incubation with sulfonamides. The LPS incubation for 30 min induced a 1.5-fold increased *AQP5* expression compared to the control condition (*p* < 0.05; Figure 6A,B). Reproducible results of blood cells from eight independent individuals showed that the pre-incubation of PBMCs with methazolamide reduced LPS-induced *AQP5* expression to control levels, while furosemide failed to reduce it and led to a more than two-fold increase in *AQP5* expression (*p* = 0.04; Figure 6A). However, after LPS incubation, methazolamide could no longer reduce *AQP5* expression to control levels, but, instead, the *AQP5* expression was increased more than two-fold (*p* = 0.004; Figure 6B).

### 2.5. Sulfonamides Do Not Reduce Cytokine Production in Macrophage-like Cells 

In a final step, we examined whether TNF-α cytokine production changes after sulfonamide incubation in LPS-treated cells. LPS incubation increased the TNF-α release of RAW 264.7-cells after 2 and 4 h of incubation (*p* = 0.0097 (2 h); *p* = 0.0046 (4 h); Figure 7), while simultaneous incubation with sulfonamides did not reduce TNF-α release (Figure 7). 

## 3. Discussion

This study elucidated potential novel therapeutic options for sepsis therapy, as it was demonstrated that the deteriorating *AQP5* mRNA expression [7] in immune cells can be downregulated by the sulfonamide methazolamide and it is also capable of reducing immune cell migration. Immunofluorescence staining indicated that incubation of REH cells with sulfonamides causes a change in AQP subcellular distribution. Pre-incubation of immune cells with methazolamide can further dampen LPS-induced *AQP5* expression, but downregulation after LPS administration is no longer possible. Furthermore, the therapeutic administration of sulfonamides after LPS exposure does not appear to alter cytokine production when considering the key-role protein TNF-α. Therefore, in a septic context, prophylaxis with methazolamide might be a promising method to modulate immune cell migration. Taken together, our results suggest that decreased AQP5 expression associated with decreased immune cell migration should be beneficial in sepsis. 

Cell migration is a process that plays a pivotal role in many physiological functions, including not only the immune response or organogenesis in the embryo but also in pathological processes, such as cancer metastasis [15]. In sepsis, proper regulation of immune cell migration is useful and essential for host defense, but dysregulated immune response is considered to be one of the most relevant pathomechanisms of sepsis [16] and an overwhelming leukocyte migration is involved in organ injury such as encephalopathy [17]. Therefore, both excessive stimulation and complete inhibition of immune cell migration should be avoided in sepsis therapy. Thus, moderate inhibition of immune cell migration by AQP5 inhibitors could be beneficial in sepsis. Our study indicates that methazolamide could be such an inhibitor. In the context of cell migration, research over the past two decades elucidated AQPs as an important regulator of many cell migration-related processes [18]. The exact role of AQP5 in cell migration is not yet fully understood, however, it has been shown that AQP5 recruits some proteins through the presence of protein-binding motifs, which are necessary for cell migration [9]. Exactly one homologous motif in the AQP5 extracellular connecting loop C (CL3) is involved in fibronectin binding and could, thereby, play a role in cell migration [9]. Another possible mechanism by which AQP5 is thought to facilitate cell migration is by mediating water influx into membrane protrusions, causing actin reorganization and the formation of lamellipodia, that provide a foundation for the cell vectored movement [18]. In the current study, we utilized the lymphocytic REH cell line for migration assay because it is a cell line with high basal AQP5 expression, which was required for the proper performance of inhibition assays. If this assumption is correct, one should have expected that AQP5 expression is decreased at the cell membrane after methazolamide incubation. This was not confirmed by our immunofluorescence staining. In addition, it must be considered that the mechanism of lamellipodia-induced migration is not predominantly described in lymphocytes [19], whereas neutrophil cells and macrophages most commonly migrate into tissues via lamellipodia formation [20,21,22]. Nevertheless, SDF1-α induced REH cell migration has been previously described [23] and our results demonstrated that AQP5 reduction in REH cells reduced the SDF1-α induced cell migration. 

However, as AQP5 also forms a pore permeable to H_2_O_2_, its potential role in sepsis must be discussed. There is evidence that H_2_O_2_ might have a causal role in the development of sepsis and toxic levels of blood H_2_O_2_ have been documented in patients with sepsis. This H_2_O_2_ toxicity can result in laboratory and clinical abnormalities observed in sepsis, including immunosuppression, bioenergetic organ failure and hypotension [24]. Hence, H_2_O_2_ influx through AQP5 channels might aggravate these processes on the cellular level. In addition, novel, more mechanistic studies indicate that H_2_O_2_ might be crucial for NLRP3 inflammasome-mediated interleukin (IL)-1β production and cell death. Although IL-1β release is dependent mainly on the apoptotic pathway, H_2_O_2_ also mediates caspase-1-dependent IL-1β production [21]. It is noteworthy to mention that it is plausible to expect cross-talk between the various cell death pathways in a complex cellular environment [25]. Furthermore, it was demonstrated that the AQP9 inhibitor RG100204 also can attenuate the activation of NF-ĸB and the expression of the NLRP3 inflammasome in the heart and kidney [26]. Again, we might conclude that AQP5 has a potential role in the activation of NF-ĸB and the function of the NLRP3 inflammasome. Further studies should examine this as it remains speculative. 

Some studies have examined the effect of methazolamide on AQP5 expression. It was demonstrated that AQPs increase cell membrane CO_2_ diffusivity, and it has been proposed that they may serve as transmembrane channels for CO_2_ and other small gas molecules. Consistent with our results, methazolamide inhibited CO_2_ exchange by 30% in buffer-perfused lungs and by 65% in blood-perfused lungs of rabbits [27]. However, it should be kept in mind that CO_2_ can also move through the phospholipid bilayer and that AQPs are certainly not needed to move this gas across the plasma membrane [28].

Our study demonstrated a strong inhibition of AQP5 by methazolamide, however, others stated that methazolamide, whose chemical structure is similar to acetazolamide, shows no significant influence on water conduction by AQP4 or AQP1 [29]. 

In contrast to methazolamide, furosemide was only capable of reducing *AQP5* mRNA expression, which caused decreased AQP5 protein in the subcellular distribution but had no effect on immune cell migration. The effects of furosemide administration on AQP5 expression seem to be different because, on the one hand, specific intracellular inhibition of human AQP1 by the diuretic drug furosemide has been demonstrated [30] and, on the other hand, furosemide treatment increased the urinary excretion of AQP2 and the activity of the renin–angiotensin–aldosterone system [31]. Hence, the regulation of AQPs by sulfonamides seems to depend on the specific AQP and possibly also on the particular cellular context. We can only speculate about the underlying mechanism. Some studies suggest that sulfonamides alter protein transcription while affecting fluid flow by a concomitant effect [32]. We can only speculate about the mechanism of sulfonamide action on AQP expression and activity. Recently, it was demonstrated that bumetanide, another sulfonamide, can restore AQP4 depolarization and also downregulate AQP4 mRNA and protein expression partially via inhibition of the ERK/MMP9 signaling pathway [33]. In addition, another sulfonamide acetazolamide can reduce the level of AQP1 protein by inhibiting the activation of the NF-κB pathway or the Wnt/β-catenin pathway. However, it seems not clear yet whether acetazolamide directly binds to AQP1 and inhibits its function by blocking the pore or only reduces AQP1 mRNA expression [34]. 

Sulfonamides provide potent antibiotics (e.g., sulfamethoxazole) and anti-inflammatory drugs, such as sulfasalazine, acting on cyclooxygenase inhibition and other pathways. As sulfonamides are a large family of synthetic antibiotics [14], there is a primary antimicrobial effect that is potentially beneficial in the treatment of sepsis. Notably, this is not the focus of this study, but additional anti-inflammatory activities or sulfonamides are also known. In detail, sulfonamide derivatives were shown to be inhibitors of caspases (e.g., caspase-1) which are key players within the intracellular inflammation-cascade [35]. Moreover, antiviral activity through inhibiting HCMV proteases has been demonstrated [36]. This might be of high relevance in sepsis, as HCMV is known to be an aggravating factor when reactivating during sepsis [37]. Even only latent HCMV might be of detrimental effect in sepsis [38], highlighting a potential benefit of anti-HCMV drugs. These entire data show a potential effect of sulfonamides during sepsis at various levels, launching scientific interest in its impact on protein expression as well. In addition, sulfonamide derivatives are commonly used diuretics and carboanhydrase inhibitors: furosemide, dorzolamide, methazolamide and acetazolamide. Their usage in sepsis is controversial. Studies examining the effect on sepsis-induced acute kidney injury (AKI) indicate that such diuretics should not be used to prevent AKI and suggest that diuretics should not be used to treat manifest AKI because the prophylactic use of furosemide to prevent AKI has been shown to be ineffective and even harmful in critical illness [39,40]. In addition, the usage of diuretics to reduce the severity of AKI once established is not evident [41,42]. Furosemide could be useful for reaching a fluid balance and reducing pulmonary edema in patients with acute lung injury [42]. A novel study recently showed that combined furosemide and aminophylline improved the urine output, fluid balance and SOFA score and reduced the ICU-, hospital- and 28-day mortality, with no worsening impact on the renal function [43]. Hence, there will be effects of these drugs on several levels. As we surely cannot elucidate all effects of sepsis, we focus here on the impact on AQP5 expression, which was shown to be important in sepsis survival. In addition, our study suggests no therapeutic effect on inflammatory response as measured by TNF-α secretion from a macrophage cell line. This is in contrast to former results showing a significant reduction in levels of TNF-α and IL-6 at a furosemide concentration of 0.5 × 10^−2^ M and a reduction in IL-8 levels at 10^−2^ M, which was comparable to that found with equivalent molar concentrations of hydrocortisone [44]. However, this concentration was five times higher than our maximum administered dose, resulting in approximately 80% death of cells. 

Studies on the effect of methazolamide in sepsis have not yet been published. To the best of our knowledge, investigations into the effects of LPS and methazolamide are limited to one study demonstrating that methazolamide mitigates lung inflammatory parameters and pathology in LPS-induced acute lung injury in mice. Here, IL-6 and monocyte chemotactic protein 1 in lung tissue and bronchoalveolar lavage fluid were decreased in mice treated with methazolamide [45]. In addition, myeloperoxidase activity, which can be used as a marker for neutrophil infiltration, was also reduced after treatment with methazolamide [45]. Since the methazolamide-treated mice had a better general state measured by physical activity and food intake and a longer survival time, methazolamide administration might be beneficial in septic patients with acute lung injury or acute respiratory distress syndrome. 

Methazolamide is a potent carbonic anhydrase inhibitor [46] and recent studies have examined the effect of carbonic anhydrase inhibitors and their usage in critical care medicine. A meta-analysis concluded that carbonic anhydrase inhibitor therapy in patients with respiratory failure and metabolic alkalosis may shorten the duration of mechanical ventilation and have beneficial effects on blood gas parameters in mechanically ventilated patients [13]. 

Taken together, the results from methazolamide and furosemide in other studies indicate that there could be an effect of these substances on the release of TNF-alpha [30,31]. However, this could not be seen in our analysis. We can only speculate about the reasons. One possible explanation could be that the substances we utilized were at lower dosages than in other studies or that the decrease in cytokine release is cell-type specific or only works in complex cellular systems in vivo. 

However, the question of why a pre-treatment with methazolamide could be effective, but a therapy after inflammatory stimuli would fail remains unclear. We can only speculate that similar mechanisms described in endotoxin tolerance might play a role. The mechanisms could be multi-level, involving receptors, signaling molecules, negative regulators and DNA-methylation as well as post-transcriptional changes, such as chromatin remodeling and microRNA regulation [47]. It has been proven that an initial LPS stimulation causes changes in gene expression and protein signaling networks. These changes could alter the strength and duration of inflammatory signaling and, ultimately, favor the activation of different transcription factor complexes upon restimulation of the cell. This might lead to changes in histone modifications, DNA methylation and chromatin remodeling machinery, which could result in an altered pattern of gene expression and the LPS tolerance phenotype [48]. However, the exact mechanisms preventing the inhibition of AQP5 expression cannot be elucidated here. It is possible that genetic or epigenetic factors may play a role here since LPS inhibits methylation at the *AQP5* promotor [49,50] and they contribute to the differential disease severity [51,52]. 

Besides sulfonamides, other modulators of aquaporins are discussed in the current literature, which is the traditional Chinese herbal drug dachengqi decoction [53], emodin [10] or hydrogen-rich saline [54], which were not considered in our study. 

The limitations of our study should be mentioned. First, we utilized a standardized cell line for our expression analysis and we cannot exclude other regulatory mechanisms in other cell lines. However, we could confirm the downregulation of *AQP5* mRNA after LPS induced AQP5 expression in human primary PBMCs. Nonetheless, the results of our cell culture experiments may be limited in translation to human physiology. To obtain more transferable results, the use of methazolamide as a prophylactic agent in sepsis should be tested in an animal model and eventually validated in a clinical trial. In addition, it has to be mentioned that the dosages used for the proliferation assay are different between the cell lines. However, a dosage of 10^−3^ M methazolamide reduced the viability of RAW264 cells, while administration of 10^−4^ M methazolamide had no effect on the viability of RAW264 cells, but reduced the viability of REH cells. Since we wanted to find out the dosage that has no effect on viability, we believe that the differences in dosage are of minor importance.

In summary, methazolamide appears to be a promising approach to sepsis prevention as it modulates immune cell migration. The knowledge about the potential benefits of this diuretic drug might, at least, drive the choice of therapy regimen in patients at risk of developing sepsis. However, our current results are only limited to in vitro studies mainly with cell culture models. Nevertheless, experiments with whole blood samples of healthy volunteers confirmed the preventive downregulation of AQP5. Animal studies and clinical trials will be necessary in the future to get a deeper view of the mechanisms and the effectiveness of methazolamide in sepsis. 

## 4. Materials and Methods

### 4.1. Cell Culture

The REH cell line was chosen after examining basal *AQP5* expression in silico with human protein atlas (https://www.proteinatlas.org/ (accessed on 16 August 2023), Version: 23.0, Executive Management Group, KTH Royal Institute of Technology, Stockholm, Sweden) analysis and qRT-PCR in different immune cell lines and comparing it with the expression of all other AQPs. The AQP5 is expressed 8.5 times higher in the REH cell line than all other AQPs, which have very low to undetectable abundances. The RAW cell line was chosen for control experiments as it is an established cell line for cytokine studies after the LPS challenge [55]. The human B cell precursor cell line REH (origin: Cell Lines Service, CLS, Eppelheim, Germany) and the murine macrophage cell line RAW 264.7 were cultured in 90% Roswell Park Memorial Institute (RPMI) 1640 supplemented with 10% heat-inactivated (h.i.) fetal calf serum (Gibco, Darmstadt, Germany) with 1% penicillin/streptomycin (Gibco, Darmstadt, Germany) at 37 °C and 5% CO_2_. Cells were maintained every three to four days by adding 5 mL of Trypsin-EDTA 0.25% (Gibco, Darmstadt, Germany) after medium removal to dissolve adhesive cells. Furthermore, PBMC samples were obtained, after the Ethics Committee’s approval (Ethics Committee of the Ruhr-University Bochum, Bochum, Germany; ref: 17-5964-BR) and written informed consent had been obtained. A volume of 80 mL EDTA blood (EDTA tubes, BD Vacutainer, Franklin Lakes, NJ, USA) was taken from eight healthy donors (five female and three male) and PBMCs were isolated using density gradient centrifugation with Ficoll-Paque (GE Healthcare, Chalfont, UK).

### 4.2. Viability Assay 

A cell viability assay was performed to identify the highest concentration of sulfonamides with no effect on cell viability. A number of 5 × 10^4^ REH cells or 5 × 10^4^ RAW 264.7 cells were seeded in 100 µL RPMI1460 + 10% h.i. + 1% P/S and cultured at 37 °C and 5% CO_2_ overnight. The following day, cells were incubated with 10^−3^ M, 10^−4^ M, 10^−5^ M, 10^−6^ M, 10^−7^ M or 10^−8^ M methazolamide, furosemide and dorzolamide (all Sigma-Aldrich, St. Louis, MO, USA) for 24 h. Afterward, viability assays were carried out using CellTiter-Blue^®^ Cell Viability Assay (Promega, Madison, WI, USA), according to the manufacturer’s instructions. 

### 4.3. AQP5 mRNA Expression Analysis 

After identification of the highest possible concentration of sulfonamides with no effect on cell viability, cell stimulation was performed for expression analysis. In order to investigate the effects of sulfonamides on *AQP5* expression, 1 × 10^6^ REH cells were cultured overnight in 4 mL RPMI 1640 + 10% h.i. + 1% P/S and incubated with 10^−5^ M, 10^−6^ M, 10^−7^ M or 10^−8^ M methazolamide, furosemide and dorzolamide. After 24 h, RNA was extracted using RNeasy Mini Kit (Qiagen, Hilden, Germany) and 1 μg RNA was used to synthesize cDNA. The following primers were used to perform real-time PCR for *AQP5*: (forward primer) 5′-CAACAACAACACAACG-3′ and (reverse primer) 5′-TAGATTCCGACAAGGT-3′, resulting in a 168 bp fragment. Primers for the housekeeping gene actin were used as described [56].

The real-time PCR reaction was performed using the GoTaq qPCR Master Mix (Promega). A cDNA dilution series for *AQP5* confirmed a PCR efficiency greater than 95%, which was comparable to the efficiency of the qPCR of actin. Relative *AQP5* mRNA expression was measured by two-step real-time PCR with actin as an internal control and calculated as 2^−^[Ct(AQP5) − Ct(β-actin)].

### 4.4. AQP Protein Expression Analysis

Regarding protein extraction, 1 × 10^6^ REH cells were cultured overnight in 4 mL RPMI 1640 + 10% h.i. + 1% P/S and incubated with 10^−5^ M methazolamide and 10^−6^ M furosemide. After 24 h incubation, the REH cells were lysed with radioimmunoprecipitation assay buffer, and proteins were extracted by shaking at 4 °C. After centrifugation, proteins were present in supernatants and could be collected by pipetting. The protein concentration was determined using a BCA Protein Assay (Pierce, Rockford, IL, USA). As a positive control for Western blotting of HEK293 cells, transfected with EX-T1015-M09 (AQP5) pReceiver-M09, protein lysates were utilized. Equal amounts of proteins of REH cell lysates were separated by sodium dodecyl sulfate–polyacrylamide gel electrophoresis (SDS-PAGE). A ROTI^®^Mark TRICOLOR ladder was utilized for protein size quantification (Carl Roth, Karlsruhe, Germany). An amount of 200 μg protein was loaded per lane on a 12% sodium dodecyl sulfate gel (SDS-PAGE) for AQP5 measurement. After separation, proteins were transferred to the nitrocellulose membrane, and an equal amount of protein was verified by Ponceau staining. Western blot analysis was performed with anti-human AQP5 antibody (G-19, sc-9890; 1:400; Santa Cruz Biotechnology), after blocking in 5% skim milk and actin antibody (clone C4; Millipore, Temecula, CA, USA). Actin was used as a loading control. Incubation with the first antibodies was performed overnight. LI-COR antibodies labeled with IR-Dys were used as a second antibody, and imaging was performed using the Odyssey imaging system (LI-COR Biosciences, Lincoln, Neb). The analysis of band intensities was performed using the online tool: https://www.freeonlinegelanalyzer.com/ (accessed on 16 August 2023). The intensities of AQP5 bands were divided through the intensities of ACTB bands. 

### 4.5. Cytospin and Immunofluorescence

After 24 h of stimulation, 2 × 10^5^ REH cells, suspended in a fully supplemented medium, were centrifuged onto glass slides by Cellspin (Tharmac, Limburg, Germany) at 1200 rpm for 5 min. The cells were fixed with 4% formaldehyde (F1635, Sigma Aldrich, Taufkirchen, Germany) for 30 min at 4 °C and washed with phosphate-buffered saline (PBS) (D8537, Sigma Aldrich).

To permeabilize fixed cells, they were treated with 0.1% (*v*/*v*) Triton-X100 (T8787, Sigma Aldrich) in PBS for 5 min at room temperature (RT). After a washing step with PBS for 5 min, further permeabilization with 0.1% (*m*/*v*) sodium dodecyl sulfate (SDS) (0183.1, Carl Roth, Karlsruhe, Germany) for 5 min was carried out, followed by three washing steps with PBS. Unspecific binding sides were blocked with Duolink^®^ blocking solution (DUP82007, Sigma Aldrich) for 30 min. The slides were incubated with a rabbit anti-human AQP5 antibody (1:50 in PBS, PA536529, Invitrogen, Waltham, MA, USA) at 4 °C overnight. Next day the slides were washed three times with TRIS-buffered saline (Carl Roth), supplemented with 0.1% Tween-20 (9127.1, Carl Roth) (TBST) for 10 min then incubated for 1 h at 37 °C with goat anti-rabbit IgG AlexaFluor^®^ 488 (1:400 in PBS, ab150077, Abcam, Cambridge, UK), followed by 3 washing steps with TBST for each 10 min. For counter stain, a few drops of SlowFade™ Gold Antifade Mountant with DAPI (S36939, Invitrogen) were used.

Fluorescence microscopy was performed with an Olympus IX51 (Olympus, Hamburg, Germany). All samples were imaged and analyzed with the same settings using Fiji ImageJ.

### 4.6. Migration Assay 

To investigate the migration of REH cells, we chose a transwell migration assay that had already been established in our group [7]. To stimulate the migration of REH cells, which represents a B-cell line, we chose SDF1-α (CXCL12) as a well-known chemoattractant for B-cell migration [57]. An amount of 2 × 10^6^ REH cells was seeded in 4 mL RPMI 1640 and incubated with methazolamide (10^−5^ M) and furosemide (10^−6^ M). After 24 h, cells were collected by centrifugation and 1 × 10^6^ cells were deposited in 500 µL RPMI 1640 into the upper compartment of a filter migration assay system containing a polycarbonate membrane filter (8 µm pore size, BD, Heidelberg, Germany). The lower compartment contained 100 ng/µL SDF-1α (Sigma-Aldrich, Taufkirchen, Germany) in 1000 µL RPMI with 0.1% BSA or control media. The cells were incubated at 37 °C with 5% CO_2_ in air for 2 h in duplicate for each sample, as described previously. The migrated cells were counted using a CellTiter-Blue^®^ Cell Viability Assay, according to the manufacturer’s instructions.

### 4.7. AQP5 Expression in Blood after Furosemide and Methazolamide Administration

Whole blood samples from healthy volunteers were collected after a statement from the ethics committee of the Ruhr University Bochum (Reg.-No: 17-5964) and receiving written informed consent. The whole blood samples were incubated with LPS (10 ng/mL), furosemide (2 × 10^−4^ M) and methazolamide (1 × 10^−4^ M) for 5 to 90 min. We differentiated between pre-incubation with LPS (therapeutic) and post-incubation with LPS (preventive) to the incubation with sulfonamides. After isolation of mononuclear cells (PBMCs), RNA isolation and reverse transcription, *AQP5* expression was quantified by real-time PCR (qPCR), as described above.

### 4.8. Cytokine Measurement of RAW 264.7 Macrophages 

An amount of 2 × 10^4^ RAW 264.7 macrophages was seeded in 100 µL RPMI 1640 and stimulated with 10^−4^ M of dorzolamide, furosemide and methazolamide in combination with LPS (1µg/mL) for 24 h. Cell culture supernatants were collected after 2, 4 and 24 h, snap-frozen and stored at −80 °C until further analysis. TNF-α ELISA (BioLegend, San Diego, CA, USA) was carried out with 10 µL of the sample and performed according to the manufacturer’s instructions. 

### 4.9. Statistical Analysis

Continuous parametric variables are presented as mean ± SEM, and their values were compared by unpaired *t*-test, paired *t*-test, Wilcoxon–Mann–Whitney tests or one- or two-way analysis of variance (ANOVA), depending on the experiment, followed by a Tukey post hoc test (Tukey multiple-comparisons test). All statistical analyses were performed using GraphPad Prism 6 (La Jolla, CA, USA). Differences were regarded as statistically significant with an a priori α error *p* < 0.05.

## 5. Conclusions

With this study, we aimed to uncover potential new therapeutic approaches in sepsis therapy or prophylaxis. As methazolamide reduces *AQP5* mRNA expression and immune cell migration, this could represent a new therapeutic strategy for sepsis treatment. Since sulfonamide drugs have relevant side effects, the dosage and timing as well as the duration of administration with the aim of inhibiting AQP-5 expression must be considered. From a clinical point of view, an early application at the onset of sepsis or maybe even in severe infections with a high risk of sepsis seems to be reasonable. Testing this in an animal model is an essential step in further examinations in this field. Hence, as a first step, an animal model such as the CLP model for sepsis should be performed to clarify whether methazolamide can be used therapeutically or only preventively. Subsequently, the effect of sulfonamides, in particular methazolamide, could be tested in a clinical, placebo-controlled double-blind study on septic patients. 

## Figures and Tables

**Figure 1 ijms-25-00610-f001:**
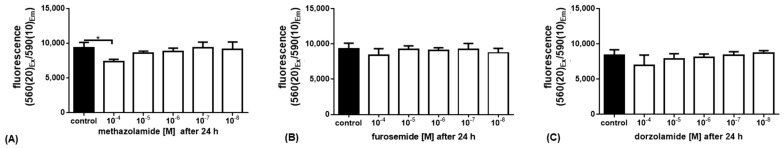
Cell viability of REH cells after incubation with different concentrations (10^−4^ to 10^−8^ M) of (**A**) methazolamide, (**B**) furosemide and (**C**) dorzolamide for 24 h (white bars, control: black bar). Viability was estimated using a CellTiter-Blue^®^ Cell Viability Assay (n = 3; * *p* < 0.05; one-way ANOVA).

**Figure 2 ijms-25-00610-f002:**
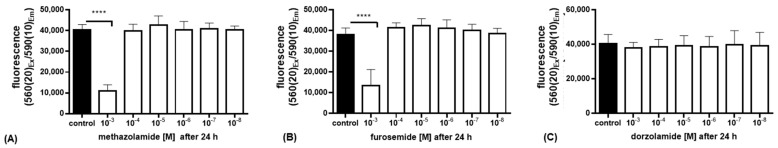
Cell viability of RAW 264.7 cells after incubation with different concentrations (10^−3^ to 10^−8^ M) of (**A**) methazolamide, (**B**) fuorosemide and (**C**) dorzolamide for 24 h (white bars, control: black bar). Viability was estimated using CellTiter-Blue^®^ Cell Viability Assay (n = 3; **** *p* < 0.0001; one-way ANOVA).

**Figure 3 ijms-25-00610-f003:**
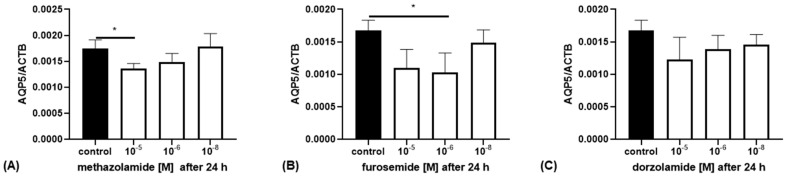
The *AQP5* mRNA expression in REH cells after incubation with different concentrations (10^−5^ to 10^−8^ M) of (**A**) methazolamide, (**B**) fuorosemide and (**C**) dorzolamide for 24 h (white bars, control: black bar). *AQP5* mRNA expression was quantified compared to β-actin (ACTB) and measured with qRT-PCR (n = 3; * *p* < 0.05; one-way ANOVA).

**Figure 4 ijms-25-00610-f004:**
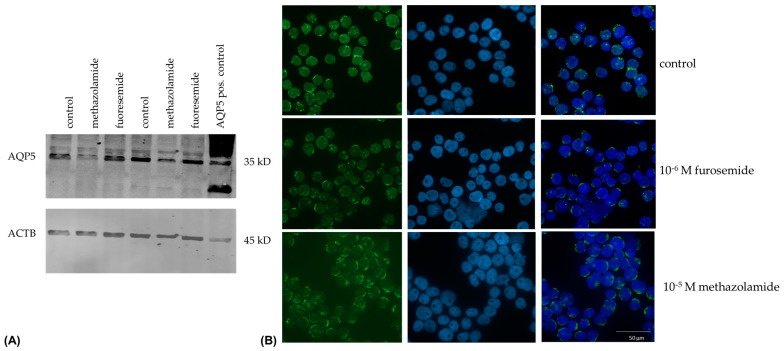
AQP5 protein expression in REH cells after incubation with 10^−5^ M methazolamide and 10^−6^ M furosemide for 24 h. (**A**) Western Blot analysis: The cells were lysed with radio immunoprecipitation assay buffer and proteins were separated by SDS-PAGE. A representative blot out of two experiments is shown. (**B**) Immunofluorescence staining after fixation with PFA, AQP5-staining (green) and counterstaining with DAPI (blue). The cell incubation, Western blotting and immunofluorescence were performed three times (n = 3).

**Figure 5 ijms-25-00610-f005:**
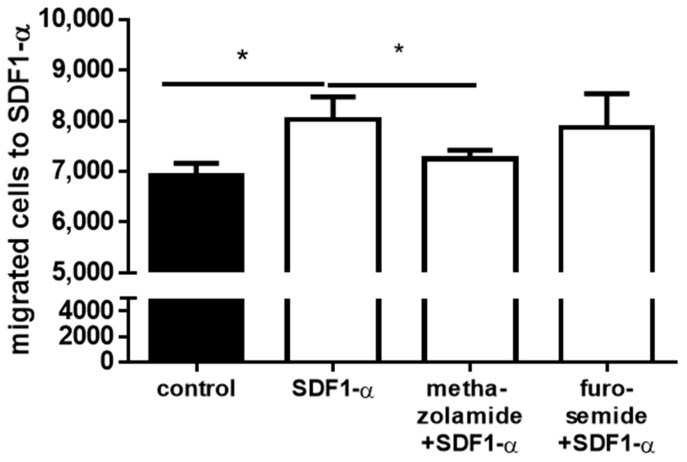
Migration assay of REH cells. Cells were preincubated with 10^−5^ M methazolamide and 10^−6^ M furosemide for 24 h. 1 × 10^6^ cells were placed in the upper compartment of a transwell insert (n = 3; * *p* < 0.05; unpaired *t*-tests) (white bars, control: black bar).

**Figure 6 ijms-25-00610-f006:**
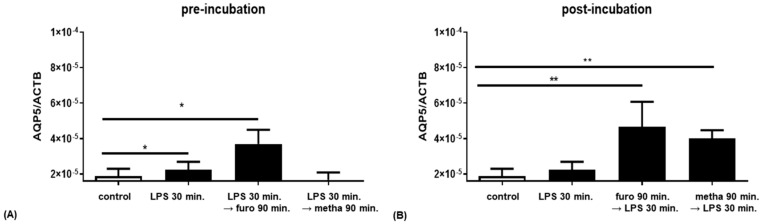
*AQP5* expression in PBMCs with furosemide (furo) (2 × 10^−4^ M) and methazolamide (metha) (10^−4^ M) pre- and post-incubation with 10 ng/mL lipopolysaccharides (LPS) (n = 8; * *p* < 0.05; ** *p* < 0.01; paired *t*-test). The → means “followed by” and indicates that the two stimulations are subsequently given after a time period of 30 min. (black bars: stimulated cells, control: white bar).

**Figure 7 ijms-25-00610-f007:**
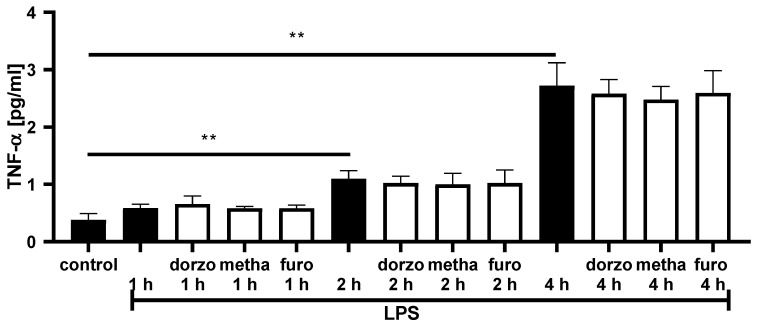
TNF-α cytokine production in RAW 264.7, stimulated with LPS (1 µg/mL) and simultaneous incubation with dorzolamide (dorzo) (10^−4^ M), methazholamide (metha) (10^−4^ M) or fuorosemide (furo) (10^−4^ M) for up to 4 h (n = 3; ** *p* < 0.01; unpaired *t*-test) (white bars: stimulated cells, control: black bar).

## Data Availability

The data presented in this study are available on request from the corresponding author. The data are not publicly available due to the containment of personal information of the probands.

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
