# Peer review of "Methazolamide Reduces the AQP5 mRNA Expression and Immune Cell Migration—A New Potential Drug in Sepsis Therapy?"

_ijms, 2024, doi:10.3390/ijms25010610_

Round 1

Reviewer 1 Report (New Reviewer)

Comments and Suggestions for Authors

The paper by Katharina Rump et al. (manuscript ijms-2775708) explores the effects of three well-known and largely used sulfonamide compounds on cell viability and aquaporin 5 (AQP5) expression at mRNA and protein level of several cell lines: a human acute lymphoblastic cell line (Reh), a mouse macrophage cell line derived from a tumor induced by the Abelson murine leukemia virus (Raw264.7), and human patient-derived peripheral blood mononuclear cells (PBMCs), as well as effects on Reh cells migration in a CXCL12 gradient and TNF-alpha secretion by Raw264.7 macrophages. The three compounds tested are furosemide, a potent loop diuretic via inhibition of the Na-K-Cl cotransporter in the ascending Henle loop, and the carbonic anhydrase inhibitors methazolamide and dorzolamide, used in primary and secondary glaucoma. The authors noticed dose-dependent decreases in AQP5 expression induced by the three drugs in Reh lymphoblasts, as well as decreases in Gram-negative bacterial endotoxin lipopolysaccharide (LPS)-induced overexpression of AQP5 in PBMCs by methazolamide only if the cells were preinchbated with the drug, inhibition of chemokine-activated Reh cells migration by the same drug, and lack of inhibitory effect of any of the three compounds on LPS-induced TNF-alpha secretion by Raw264.7 macrophages. The study was well conceived and planned, the experiments were accurately performed and correctly interpreted, leading to valuable conclusions regarding the potentia clinical usefulness of these medications in septic shock, therefore I consider the manuscript suitable for publication in International Journal of Molecular Science. It is worth mentioning that furosemide and acetazolamide are known inhibitors of aquaporins 1 and 4 (Abir-Awan M et al. 2019 IJMS 20:1589); furosemide applied on the internal side of a cut-open Xenopus laevis oocye injected with human AQP1 mRNA impaired osmotic swelling (Ozu M et al. 2011 Eur J Biophys 40:737–746). US patent 2015/0224108 A1 describes synthetic derivatives of the loop diuretics furosemide and bumetanide and their potential use as aquaporins inhibitors. Other patents listed in the review of Abir-Awan M et al. describe various AQP inhibitors derived from loop diuretics, phenylbenzamides, as well as other methods of aquaporins inhibition such as siRNA silencing or monoclonal antibodies (aquaporumab - AQP4-specific IgG MoAb). Adamzik M et al. 2011 Anesthesiology 114:912–7 show that the 1364A/C polymorphism in the AQP5 promoter region is associated with improved 30-days survival in severe sepsis, strengthening the conclusions of the current manuscript. However, the important diuretic and blood pressure-reducing effects of loop diuretics like furosemide are somehow in conflict with the therapy principles in severe sepsis, where blood pressure has too be maintained by intravenous fluids perfusion and vasopressors such as norepinephrine. I have encountered ocassionaly furosemide administration in patients with severe septic shock as a heroic or rather desperate measure to preserve diuresis in conditions of dropped blood pressure, but the results were not conclusive and I am not aware of a systematic study on this topic. It may be the case, like stated by the authors, that methazolamide could be effective only if adminstered prophylactically prior to a major endotoxin release and subsequent shock.

Some suggested minor corrections are inserted in the pdf version of the manuscript sent along with this review.

Comments on the Quality of English Language

Some suggested minor corrections are inserted in the pdf version of the manuscript sent along with this review.

Author Response

We would like to thank the reviewer for the detailed feedback on our study and appreciate the positive response. We have incorporated the suggested changes into the manuscript and feel that it has been greatly improved as a result. Thank you again for the time and effort you have put into improving our manuscript. 

Reviewer 2 Report (New Reviewer)

Comments and Suggestions for Authors

Dear editors and authors:      

        It is a great honor and pleasure for me to be invited as the reviewer for this important work entitled “Sulfonamides reduce the AQP5 expression and immune cell migration – new potential drugs in sepsis therapy?”. Katharina Rump and co-authors investigated the application of Sulfonamides on the inhibition of AQP5 expression and immune cell recruitment. This study topic is novel and interesting, attributing to corresponding author Matthias Unterberg’s long-term efforts and contributions in this scientific field. I have a number of comments concerning this study:

1.     There are myriads of Sulfonamide derivatives that the title “Sulfonamides” should be replaced by “Methazolamide” to highlight the scope of the study.

2.     The conclusion in the abstract is weak that should be rephrased based on the study results: sepsis therapy or prophylaxis use.

3.     Sulfonamides had been the first widely used antibacterial drugs since 1935. Nowadays sulfonamides alone are hardly the drugs of first choice because of the availability of more effective antibiotics and the high prevalence of resistance. From the perspective of clinicians, how could authors conclude that methazolamide appears to be a “promising approach” in sepsis prevention despite of its modulation on immune cell migration?

4.     The section of limitation should be provided.

5.     Did the results support the title and conclusion?

6.     In light of the conclusion in line 293, what is the timing/ indication of drug administration for sepsis prevention?

Author Response

Thank you very much for reviewing our manuscript. We give a detailed response in the word document. 

Reviewer 3 Report (New Reviewer)

Comments and Suggestions for Authors
  1. Abstract and Keywords:

o   on percentage in AQP5 expression and cell migration in REH cells.

o   Keywords are well-chosen but consider adding "immune cell migration in sepsis" or "AQP5 inhibition."

  1. Introduction:

o   Provide more detailed statistics on sepsis incidence and its impact on healthcare systems. Reference recent breakthroughs in sepsis treatment and the role of AQP5.

o   Expand on the potential of sulfonamides as therapeutic agents in sepsis, citing relevant studies.

  1. Methods:

o   Clarify the choice of cell lines (REH and RAW 264.7) for the study. Detail the concentrations of sulfonamides used and their justification.

o   Offer more information about the migration assay, including the rationale behind the choice of SDF1-α as a chemoattractant.

  1. Results:

o   Present more detailed statistical analysis results for cell viability assays and AQP5 expression studies. For instance, specify the percentage change in cell viability at different concentrations of methazolamide.

o   In the section on immune cell migration, describe the quantitative effects of sulfonamides more precisely.

  1. Discussion:

o   Discuss the implications of AQP5 inhibition on immune cell migration in the context of sepsis therapy, referencing specific findings from your study.

o   Explore the potential mechanisms behind sulfonamides' effects on AQP5 expression and immune cell migration, relating them to existing literature.

  1. Conclusions:

o   Highlight the study's implications for developing new therapeutic strategies in sepsis, focusing on the role of AQP5 and sulfonamides.

o   Suggest potential clinical trials or further research needed to validate the findings.

  1. Figures and Tables:

o   Ensure that figures clearly represent the experimental data. For example, graphs showing AQP5 expression should be easy to interpret.

  1. References:

o   Update references to include the most recent studies in the field of AQP5 and sepsis therapy.

o   Consider adding more recent references to support statements, especially in rapidly evolving areas of the field. Where possible, include recent studies to demonstrate the manuscript's alignment with current research trends. In particular, consider including additional references to support the discussion and to provide context to the study’s findings. I suggest adding data related to recent bulk transcriptomics studies which could represent a strong substrate to enforce the role of described molecular mechanisms, such as the recent PMID: 36490268 and PMID: 27737651.

  1. Ethical Considerations:

o   Confirm that the manuscript includes appropriate ethical statements related to the use of human and animal cells.

Comments on the Quality of English Language

The English should be improved.

Author Response

Thank you very much for reviewing our manuscript. We give a detailed response in the word document. 

Round 2

Reviewer 1 Report (New Reviewer)

Comments and Suggestions for Authors

The revised version of the manuscript by Katharina Rump et al. (ijms-2775708-v2) is improved relative to the initial version and may be published in IJMS. 

Comments on the Quality of English Language

I suggest changing the term "depict", which is often used in the text but inadequately from my point of view (because it means to paint or to describe) with "represent" (e.g. on page 6 line 190: "methazolamide could depict such an inhibitor" should be replaced with "methazolamide could represent such an inhibitor"). The same replacement should be performed elsewhere in the text. There are still a few spelling errors, which could be also corrected within the proofs.

Author Response

Dear Reviever, 

Thank you very much once again for correcting our manuscript. 
We have replaced the word in question in the text and also checked the spelling once again. 

This manuscript is a resubmission of an earlier submission. The following is a list of the peer review reports and author responses from that submission.

Round 1

Reviewer 1 Report

Comments and Suggestions for Authors

The work in question deals with the involvement of aquaporins (AQPs) in sepsis and their potential pharmacological relevance as drug targets. In particular, the authors test some sulfonamides and their derivatives on the expression and potential role of AQP5 in the cell migration of immune cell lines REH and RAW 264.7 and human PBMCs. Based on their results the authors conclude that methazolamide, a sulfonamide derivative reported to inhibit carbonic anhydrase in septic patients, reduces the AQP5 expression and migration of immune cells. They also conclude that the sulfonamide is effective when administered before inducing inflammation with LPS. The manuscript has serious flaws. In addition to having been conducted on the basis of a non-rigorous experimental design, the data obtained are interpreted in a purely speculative manner without considering many contingent aspects starting from the fact that sulfonamides are not compounds with specific inhibitory actions up to the failure to consider that lymphocytes and monocytes/macrophages used in the study express other aquaporins that have been reported to have a role in cell migration that have been completely neglected. The work is poorly written as well as the English. There are countless proofreading problems.

1) In both the REH and RAW 264.7 the authors point to AQP5 without considering that lymphocytes and macrophages also express other AQPs and do not provide any evidence demonstrating that the compounds used do not affect the other AQPs either in terms of expression or in terms of function (i.e., cell migration). This is an absolute aspect to be addressed without which no conclusion can be drawn referring to AQP5 alone. Moreover, as stated by the authors themselves, sulfonamides are not to be considered specific inhibitors.

2) The profile of Western blotting as it has been made and described is not at all convincing. Not to mention that the authors load 200 µg of lysate proteins, an abnormal quantity for a Western blotting, the profile shown does not indicate either the size of the immunoreactive bands shown or what the two bands correspond to, one lower stronger and one upper less strong. A positive and a negative control is missing, however. In order to make functional speculations, the authors would have to do confocal immunofluorescence both to demonstrate that what they are seeing as AQP5 is in the plasma membrane and to confirm the differences in expression they say they see.

3) In addition to being an orthodox aquaporin AQP5 is also a peroxiporin. In addition to specifying this in the introduction (lines 36-37), the authors should bear this peculiarity in mind in the Discussion and, more generally, in the entire work.

4) In reporting the data, the absolute numbers of the values ​​of each condition measured, nor the standard errors, are not indicated. The differences are not even indicated in percentage terms. Added to this is the fact that differences that are really minimal and in some ways insignificant are given as important differences. This is an aspect that is not taken into account in the discussion of the data.

5) The way in which at the end of the introduction the authors indicate the hypotheses investigated with the work is rather hermetic and not so clear what they intend to do. It should be rewritten.

6) The manuscript is chock full of proofreading issues that need to be fixed. Just to name a few, but there are so many, the fact of reporting numbers where the comma is in place of the point (e.g., p-values ​​in the figures), lack of lettering on the panels (a, b, c, etc.), asterisk numbers in the legends of the figures that are not found in the panels of the same figures, etc.).

7) In figures 1 and 2 the authors indicate the terminology "viable cells" but claim to measure cell proliferation. The two concepts are not exactly the same and therefore perhaps it would be more correct to use only one of the terminologies in the graphs.

8) The discussion is too speculative. If on the one hand they speak of the movement of CO2 through the AQPs (a very debated aspect), on the other they do not mention several works on AQPs of the cells of the immune system where elements that could be useful for the discussion have been reported. The authors should point out that CO2 moves very well through the phospholipid bilayer and that AQPs are certainly not needed to move this gas across the plasma membrane.

9) The first part of the Discussion repeats what was said in the Introduction and can therefore be eliminated.

10) Line 102. “A representative blot out of two experiments is shown (n=3).”: two or three experiments? If they are two they are too few and three are right on the edge.

11) The manuscript is written in poor English.

Comments on the Quality of English Language English needs to be greatly improved.

Reviewer 2 Report

Comments and Suggestions for Authors

The author attempted to explore how sulfonamides reduce AQP5 expression under various conditions. The impact of sulfonamides on AQP5 expression and immune cell migration was examined in cell lines REH and RAW 264.7 using qPCR, Western Blot, and migration assays. However, the manuscript is replete with typos and inconsistencies. The study design and duration of treatments are confusing and unclear. Additionally, all materials and methods require proper citation.

    The results of MTS need to be checked; the O.D. value of MTS cannot be greater than 3. Please check the accuracy. 

    There is a typo in figure 2's legend. Several typos in figure legend, please revised it. 

    Reference 13 does not provide any information about the migration assay's methodology. Please add the related citation for these methods.

    If there’s 3 dependent studies in figure4, please added the statistical analysis’s results. 

    Figure 6’s X axis, please check the accuracy. The description is confused and the statistical analysis is totally lack of scientific background, you can’t compare two groups with more than one factors. furo 90 min. -> LPS 30 min could only compared with LPS induced group. Author did not label any subfigure instruction. Figure 6’s description did not present any scientific based results. 

    The dosage and duration of treatment are unclear and should be specified. 

        The proliferation dosage is different between 2 cell line.

        Why choose 24H for proliferation, but used 2H or 4H for cytokine analysis.

    If author addressed the modulation of immune cell, why there’s no significant changes in TNF-a, Author should add in the discussion. 

    In material and methods used Relative AQP1 and AQP5 mRNA expres-294 sion was measured by two-step real-time PCR with actin as internal control and calculated 295 as 2−[Ct(AQP5) −Ct(β-actin)]. There’s no AQP1’s results and did not show the sequence of actin.

    The reasons of choosing methazolamide furosemide dorzolamide should be described carefully in the introduction. Dorzolamide did not alter the AQP5 mRNA level, how about its protein expression. Author should also do the protein expression. 

    Please check the formation of manuscript 

Comments on the Quality of English Language

The author attempted to explore how sulfonamides reduce AQP5 expression under various conditions. The impact of sulfonamides on AQP5 expression and immune cell migration was examined in cell lines REH and RAW 264.7 using qPCR, Western Blot, and migration assays. However, the manuscript is replete with typos and inconsistencies. The study design and duration of treatments are confusing and unclear. Additionally, all materials and methods require proper citation.

    The results of MTS need to be checked; the O.D. value of MTS cannot be greater than 3. Please check the accuracy. 

    There is a typo in figure 2's legend. Several typos in figure legend, please revised it. 

    Reference 13 does not provide any information about the migration assay's methodology. Please add the related citation for these methods.

    If there’s 3 dependent studies in figure4, please added the statistical analysis’s results. 

    Figure 6’s X axis, please check the accuracy. The description is confused and the statistical analysis is totally lack of scientific background, you can’t compare two groups with more than one factors. furo 90 min. -> LPS 30 min could only compared with LPS induced group. Author did not label any subfigure instruction. Figure 6’s description did not present any scientific based results. 

    The dosage and duration of treatment are unclear and should be specified. 

        The proliferation dosage is different between 2 cell line.

        Why choose 24H for proliferation, but used 2H or 4H for cytokine analysis.

    If author addressed the modulation of immune cell, why there’s no significant changes in TNF-a, Author should add in the discussion. 

    In material and methods used Relative AQP1 and AQP5 mRNA expres-294 sion was measured by two-step real-time PCR with actin as internal control and calculated 295 as 2−[Ct(AQP5) −Ct(β-actin)]. There’s no AQP1’s results and did not show the sequence of actin.

    The reasons of choosing methazolamide furosemide dorzolamide should be described carefully in the introduction. Dorzolamide did not alter the AQP5 mRNA level, how about its protein expression. Author should also do the protein expression. 

    Please check the formation of manuscript 

Reviewer 3 Report

Comments and Suggestions for Authors

The authors examined the effects of sulfonamides on AQP5 expression and immune cell migration, and they found that sulfonamudes downregulate the expression of AQP5 on immune cell lines, suggesting that they might serve as a potential therapeutic approach in sepsis prevention, although more research works need to be done in the future.

The manuscript was well-written, and the conclusion was supported by the experimental data. I only have some minor considerations.

1. Regarding figures 1 and 2, original cell imagines with different treatment should be provided.

2. Line 78, after Figure, the "c" should be capital.

3. Line 129, after LPS-, the blank space should be deleted. 

Round 2

Reviewer 1 Report

Comments and Suggestions for Authors

The Western blotting shown in figure 4 remains unconvincing (the size of the bands shown is not clear also because the size of the few bands of the ladder is not shown in the figure inserted in the replies to my comments; moreover, no ladder is shown in the figure inserted in the manuscript) and, even more so, the authors should demonstrate by immunofluorescence (conventional immunofluorescence may be enough if confocal cannot be done by the authors) that AQP5 is actually expressed and is inserted in the plasma membrane of REH cells. Doing this under the various experimental conditions with the inhibitors/control would also lend greater strength to the data.

When writing “…In addition, novel, more mechanistic studies indicate that H2O2 might be crucial for NLRP3 inflammasome-mediated interleukin (IL)-1β production and cell death….” The authors should cite the study/ies where the crucial role for hydrogen peroxide for NLRP3 inflammasome-mediated interleukin (IL)-1β production and cell death has been demonstrated.

The work cited in reference 7) is not the one where it has been demonstrated that the AQP9 inhibitor RG100204 also can attenuate the activation of NF-ĸB and the expression of the NLRP3 inflammasome in heart and kidney. Likely, the authors mean the work by Mohammad et al. (doi 10.3389/fimmu.2022.900906 ). This should be corrected in the reference list. The review by D’Agostino et al is worth and should be mentioned elsewhere in the text (Introduction and/or Discussion). I believe there is a problem in the reference list because the old reference 7) (D’Agostino et al) is still present in the corrected version of the manuscript while the new reference 7) (Mohammad et al.) is among the references listed in the replies to comments.  The problem needs to be fixed.

The hypothetical mechanism through which AQP5 would intervene in cell migration is not discussed, which could instead be very useful to the reader who wants to understand why AQP5 would be important in sepsis.

The way in which at the end of the Introduction the authors indicate the hypotheses investigated with the work is still not so clear and poorly written as well as not listed well.

There are still unsolved proofreading problems (i.e., APQ instead of AQP; AQP-5 instead of AQP5).

Comments on the Quality of English Language English has improved but is still not at its best.

Reviewer 2 Report

Comments and Suggestions for Authors Thanks for the clarifications and the explanations. The author have addressed all my questions and concerns.